# Assessment of the Sedia HIV Self-Test Device: Usability and Performance in the Hands of Untrained Users in Johannesburg, South Africa

**DOI:** 10.3390/diagnostics11101816

**Published:** 2021-10-01

**Authors:** Mohammed Majam, Naleni Rhagnath, Vanessa Msolomba, Leanne Singh, Michael S. Urdea, Samanta Tresha Lalla-Edward

**Affiliations:** 1Ezintsha, Faculty of Health Sciences, University of the Witwatersrand, Johannesburg 2193, South Africa; mmajam@ezintsha.org (M.M.); nrhagnath@ezintsha.org (N.R.); vmsolomba@ezintsha.org (V.M.); thewritepages@gmail.com (L.S.); 2Halteres Associates, 2010 Crow Canyon Place, Suite 100, San Ramon, CA 94583, USA; murdea@halteresassociates.com

**Keywords:** HIV testing, lay user evaluation, sensitivity, specificity, point-of-care

## Abstract

The prevalence of HIV across South Africa places a strain on testing facilities. The use of HIV self-testing (HIVST) devices has been identified as a strategy to ease the burden on these facilities. The usability and performance of the Asante HIV−1/2 Oral Self-Test (Asante) (Sedia Biosciences, Portland OR, USA) device by novice users was assessed and reported on, to inform for the implementation of such devices in South Africa and elsewhere. Convenience sampling was used. Participants used the Asante HIVST device and recorded their interpretation of their results. Participants’ interpretations were compared with those of trained professionals and, thereafter, verified using the rapid diagnostic testing algorithm. Out of the eligible participants, 410 of the 524 (78.2%) were between the ages of 18–35. The usability assessment indicates that 100% of participants used the HIVST device’s information leaflet. However, 19/524 (3.6%) of participants who yielded an invalid result due to critical errors were excluded from the primary efficacy analysis. The average usability score was 98.1%. The sensitivity and specificity results were, 94.7% and 99.8%, respectively. This study shows that the Asante HIV self-test, and similar devices, can be valuable in providing convenient HIV self-testing and immediately available results. To accommodate a greater number of inexperienced users, the instructions may need to be revised.

## 1. Introduction

To encourage more people to become aware of their HIV status, the use of HIV self-testing (HIVST) as an alternate testing strategy has seen an increase in momentum, globally [1,2]. In December 2016, the World Health Organization (WHO) released its normative guidance on HIVST [3] to regulate the manufacture and use of HIVST kits. Guided by the WHO requirements in the Technical Specification Series (TSS), The Bill and Melinda Gates Foundation funded the HIV Self-Testing Assessments and Research (HSTAR) programme, “Bringing high quality, WHO-approved HIVST to the commercial market in South Africa” in January 2016 [4]. In March 2018, the South African National Department of Health [5], in accordance with the WHO’s pre-qualification standards (WHO-PQ) [6], released its guidelines, which, like WHO, stressed the importance of using prequalified products in its national implementation programmes [7].

The HSTAR programme, developed and implemented by Ezintsha, sought to address the requirements of the WHO PQ TSS, specifically, the requirements of Part 3. These evaluations served as independent assessments of test kits that manufacturers would be able to use as part of their dossier of evidence of test usability and performance [8]. The principle focus of these assessments was to evaluate whether untrained users in Johannesburg, which has a high HIV prevalence, can correctly perform the test and interpret their results [9].

The HSTAR [10] approach that was initially adopted, employed a multi-level design for the assessment of HIVST product performance, based on Food and Drug Administration and Conformitè Europëenne (CE) mark standards for existing HIVST products at the time, as well as specific needs for self-testing of in-country target populations. The specific HSTAR programme benchmarks included: (1) developing a set of practical, robust HIVST evaluation standards and protocols for product-naïve, untrained laypersons; (2) providing a vehicle for independent and unbiased assessment of HIVST product performance; and (3) producing data that can be presented for WHO-PQ by the respective device manufacturers.

One such product, the Asante HIV−1/2 Oral Self-Test (Asante) (Sedia Biosciences, Portland, OR, USA) HIVST device, was assessed by evaluating the ability of untrained users to obtain an accurate HIV test result using the device. Additionally, the following were scrutinized: the efficacy of device usage with regards to the comprehension and successful completion of the critical steps as per the instructions for use (IFU), including key messaging and labelling. In this paper we present the results of this evaluation which employed a multi-levelled approach.

## 2. Materials and Methods

### 2.1. Study Design

A cross-sectional study was conducted assessing the performance of the self-test by untrained users. An untrained user is defined as an individual who is not trained in the performance of rapid test diagnostics, or a lay user who has not been provided with any training, support, or guidance on the tests performance other than what is available on the product’s instructions for use. Under controlled conditions, trained professionals provided participants with a HIVST package to self-test. They observed the untrained users’ conduct of the test, recorded the participants’ engagement with the device as well as the participants’ interpretation of their HIVST result. The trained observer recorded successful completion of each of the process steps as specified in the IFU, and noted any errors, inconsistencies and omissions. Interviews with participants were conducted to ascertain participants’ experience of using the test. Thereafter, the professional user would conduct confirmatory testing by way of professional use RDTs (RDT 1: First Response™ HIV−1/HIV−2 WB; Product code DD/138, Premier Medical Corporation Ltd., Kachigam, India; RDT 2: ADVANCED QUALITY™ Rapid HIV Test—Product code ITP02002-TC40, InTec Products Inc., Xiamen, China) and have a sample drawn for the laboratory gold standard HIV−1 ELISA test, performed on the Abbott Architect 1000SR HIV Ag/Ab combination (Abbott, Green Oaks, IL, USA). These results were recorded separately. The self-testers’ interpretation of their result was then compared to the professional users’ interpretation of the HIV self-test result, the confirmatory RDTs and the HIV ELISA.

### 2.2. Product

The Asante HIVST kit was used. The kit comprised of a sealed box in which the following elements were encased: an instruction leaflet, a sample buffer tube, a test strip in its own wrapper and an oral swab in its own wrapper. Printed on the box itself were instructions and contents descriptions.

### 2.3. Study Site

The investigation was conducted at the Ezintsha Clinical Research Centre in Hillbrow, Johannesburg. The participating site complied with all local government requirements for HIV testing and reporting. Sample collection and testing for the study were performed by professional nurses; other relevant study staff were trained and deemed competent in the study procedures, such as, accurate use of data collection tools, performance and usage of the HIV self-test and the instructions sheet, before the official commencement of the assessment.

### 2.4. Study Participant Recruitment

The study was open to the general population over the age of 18 who met all the inclusion criteria, irrespective of race, gender, ethnicity, or sexual orientation. The criteria comprised of prospective participants being 18 years or older, who could speak and read English; they confirmed their level of education, demonstrated comprehension of the consent form before signing it, were able to complete the required testing on the allocated testing day and agreed to provide an accurate medical history. Prospective participants who were excluded were those who: did not meet the inclusion criteria; knew they were HIV positive; had received an experimental HIV vaccine; were currently on a pre-exposure prophylaxis regimen or on any antiretroviral therapy; had participated in any prior or concurrent trial of HIV self-tests; were practicing medical healthcare professionals (doctor, nurse or HIV counsellor who performs HIV testing with rapid tests); had used a rapid diagnostic test (RDT) for self-testing previously; were unwilling to use the biometric enrolment system; or could not provide an identity number or identity documentation for the biometric enrolment. Furthermore, anyone who displayed any condition or behaviour which, in the opinion of the facilitator, would jeopardise the accuracy of the study (e.g., intoxication or having forgotten their reading glasses thus being unable to see properly) was excluded. Enrolled participants were compensated R150 for their time.

### 2.5. Sample Size and Sampling

Convenience sampling with consecutive recruitment was initiated until 524 unassisted HIV self-test users were obtained. This sample was enough to meet WHO-PQ technical standard specification requirements for a high prevalence setting [8].

The primary efficacy analysis was calculated using 503 participants. Nineteen users who made critical errors in the testing process were excluded from the primary efficacy analysis. A critical error is one that has a material impact on the determination of an accurate result. Additionally, two participants were excluded; one for testing positive for p24 Ag, and the other for refusing to answer specific questions about the start of their HIV treatment (exclusion criteria). Sample sizes throughout the study steps are reflected in Figure 1.

### 2.6. Study Procedures

Each participant was prescreened against the recruitment criteria. A biometric system was used to prevent multiple enrolment, and eligible participants were provided with a unique participant identification number. Before obtaining participants’ voluntary written consent, nurse administrators explained and ensured participants’ understanding of the content and process requirements. Thereafter, a 5 mL blood sample was drawn from each participant for a HIV ELISA laboratory test. Participants proceeded to private rooms accompanied by an observer who instructed the participant on the process. The participant was handed a HIVST package and then used it to perform the self-test procedure. On completion, the participant’s interpretation of their self-test was recorded, followed by the trained nurse’s independent recording of the participant’s HIVST result. The observer nurse then interviewed the participant, documented relevant data and performed the HIV test using the national testing algorithm (Figure 2).

### 2.7. Data Verification

Study questionnaire information was captured in real time by data capturers, 10% of which were checked at the end of each day by the quality assurance officer, using the source documentation for verification. Errors were reported to the data capturers, comprehensive corrective procedures were executed, and preventative action plans were installed when required. All source and electronic documentation were stored in restricted access-controlled cabinets to ensure security, quality maintenance and confidentiality.

### 2.8. Data Analysis

To provide an immediate, confirmed diagnosis, a nurse used the South African standard HIV testing algorithm [11] to verify participants’ self-reported results in their presence. A laboratory fourth-generation ELISA (EIA) run on Architect 1000SR was used to confirm all test results, irrespective of the algorithm test result. In cases of discordant professional test results, the EIA was used for diagnosis. The sensitivity and specificity of the HIV self-test (the participant self-reported result) was calculated relative to the fourth-generation EIA.

Usability was calculated using the average of the correctly performed steps 1 to 12 of the process and included the time used to read the process steps. Usability was calculated on 524 participants, which included the participants who had been excluded from the primary efficacy analysis.

The primary efficacy analysis included the calculation of clinical sensitivity and specificity of study participants’ results with the HIVST versus the fourth-generation EIA test results. A secondary efficacy analysis compared the HIVST results to the result obtained by the confirmatory rapid test algorithm. The proportion of study participants’ interpretations of their self-test, which was confirmed by the confirmatory test algorithm, and thereby considered “true”, was ascertained following the determination of the study participant’s data inclusion in either the sensitivity or the specificity analysis as shown in Figure 3 [12].

## 3. Results

### 3.1. Demographics

Table 1 is a representation of the demographic profile of participants. Most participants were female (276/524; 52.7%). There was diversity in the age, nationality, educational level and employment status of the participants. The majority of the participants were between the ages of 18–35 (410/524; 42,2%), and most participants were South African (475/524; 90.6%). While only 33/524 (6.3%) had primary education or less, the bulk of the participants had completed secondary schooling (310/524; 59.2%) and a further 139/524 (26.5%) were tertiary educated. The majority were unemployed (404/524; 77.1), while 87/524 (16.6%) were employed, and 33/524 (6.3%) were students.

### 3.2. Usability Assessment

Steps 1–14 of the process, represented in Table 2, were used in the usability calculation. This calculation was based on the engagement of 524 participants and included the 21 participants who were originally excluded from the primary efficacy analysis. Critical steps refer to those steps that directly impact the quality of the results, such as failure to add a sample to the device, not using the developer buffer or not collecting the required quantity for the sample. Non-critical steps are those that do not invalidate results, such as opening the package incorrectly.

Every study participant read or used the information sheet. While less than 1% had trouble with removing the packaging from the box and the package contents from their wrappers, 2.7% found it difficult to place the tube in its tube box. The majority of participants were able to follow the oral swab instructions successfully, with the highest numbers of inappropriate techniques occurring at the following steps: holding the tube in the box and pressing the swab firmly against the tube opening (2.9%), and the method for collecting the swabbed sample and disposing of the swab (3.2%). While almost all participants (99.8%) successfully opened the test strip packaging and removed the strip, 4.4% demonstrated an inability to correctly place the test strip in the tube. Just over half the participants’ preferred language was English (54.2%), and most appeared calm while performing the test (92%).

The average usability score was 98.1%, with an average score of 97.0% for the successful performance of the critical steps.

### 3.3. Primary Efficacy Assessment

Of the 524 participants that were enrolled, 20 (4.8%) made critical errors in the testing process. Of those who made a critical error, 19/20 (95%) had identified their errors and were excluded from the primary efficacy analysis set. Of these, most (10/19, 52.6%) experienced difficulties understanding what the test strip was to be used for, how to use it correctly and/or inserted the test strip upside down into the tube. Another common error was not being able to use the correct technique during the oral swabbing step. Overall, only 2/19 (10.5%) demonstrated key errors in the execution of most of the steps. The results of all 20 who had not correctly followed the process steps as indicated in the IFU were reported by the observer as other. One of the participants interpreted an invalid result as negative. The primary efficacy analysis was, therefore, measured against 503.

### 3.4. Performance Evaluation

Of the 524 participants, 503 (95.9%) succeeded in completing the HIVST independently and were included in the test performance calculation. The results of 19 (3.6%) participants who made errors executing the self-test as well as one participant who had incorrectly read the result as negative, were excluded from clinical sensitivity and specificity. One participant that tested p24 Ag positive and one that refused to answer study related questions regarding ART initiation were also excluded from the primary efficacy analysis.

Ultimately, 72/503 (15.2%) true positive results, as well as 426/503 (83.2%) true negative results were identified by both self-testers and ELISA. Furthermore, 1/524 (0.2%) false positive interpretation was reached by both the self-test and ELISA, while HIVST identified 4/524 (1.2%) false negatives.

Table 3 and Table 4 contain device performance results. This yielded a 94.7% sensitivity and a 99.8% specificity for the participants’ interpretation of the self-test against the ELISA result, and a 97.4% sensitivity and 99.8% specificity for professional user interpretation of the self-test against the ELISA (Table 5).

The RDT algorithm diagnosed 1/401 (0.2%) further positive result and 1/401 (0.2%) further negative result.

### 3.5. Post-Test Questionnaire

For questions 1–5, which focussed on the ease of reading the instructions and using the HIVST, 522/524 (99.6%) responded favourably to the self-test procedure instructions. However, pictures 1,5,6,7,8 and 9 were identified as confusing. Questions 6–9 probed the participant’s intended post-HIVST behaviour after learning of their results. At least 481/524 (91.8%) indicated that they would visit the clinic in the instance of being unsure of their result or on receiving a positive result. A smaller percentage indicated clinic visits in the event of an invalid result. Of those who received a negative result, 522/524 (99.6%) participants opted for condomisation. Questions 10–12 gave attention to participants’ opinions on future usability of the HIVST. Of those who participated, over 509/524 (97.1%) of participants would use the Asante HIVST again and/or recommend it to others. Only 34% (6.5%) chose being tested in a clinic over being tested at home (Table 6).

## 4. Discussion

In this study, the results strongly indicate that HIVST is a feasible method for initial HIV screening, with regards to self-testers being able to follow self-test kit packaging instructions through to successful completion, as well as the self-test kit providing reliable results.

These findings add validation to previous studies whose conclusions support the use of HIVST for providing users with results that have high accuracy in sensitivity as well as specificity [12,13]. Similar to this study, which yielded an average sensitivity score of 94.7% and average specificity score of 99.8%, a 2016 study in a rural Limpopo Province (South Africa) area revealed specificity and sensitivity scores of 99.02% and 100%, respectively, amongst untrained users [14]. In a related Singaporean study [15], a cross-sectional investigation reported a sensitivity of 97.4% and specificity of 99.9%. All the HIVST devices in the investigations were oral ones, and in each investigation, HIVST interpretations were compared to blood sample tests.

South Africa has subscribed to the UNAIDS goal to end the AIDS epidemic by 2030 [16]. To attain this goal, vast numbers of the population need to undergo at least initial, fairly accurate testing quickly. Additionally, there needs to be consistent availability of testing kits for all contexts. HIVST uptake suggests that devices such as the Asante HIVST can be extremely beneficial, not only in reaching lay users and providing them with prompt, accurate results, but also adds another option for an oral fluid test to the market (already known to be the most acceptable type of HIVST [17,18]).

However, 4.8% of users did commit errors related to the comprehension of package instructions, rendering, as expected, the HIVST device less than 100% user-friendly. This suggests that further iterations of the IFU are required to improve overall usability and to minimize user errors [19]. It is very possible that the level of cognition and/or literacy level of the participants affected the extent to which they could engage with the HIVST kit [20]. This view is further shared in a recent study by Simwinga et al. [21], where it was observed that those participants who had more advanced cognitive levels fared better with successful use of an HIVST. Furthermore, while the 0.2% error in specificity may be small, it does pose a threat to the responsible behaviour of users post-test; where those who obtain a false result may subsequently behave in ways that endanger themselves or others. Young HIVST users in one study [22] reported similar concerns, citing possible suicidal behaviour of those who received HIV positive results, and this possible outcome, as well as engaging in violence towards others, was seen as a possible result in another investigation [23]. Johnson et al. [24] expound on additional consequences of misdiagnoses, as they relate to social and relationship-related repercussions.

Self-tests for sexually transmitted infections other than HIV, have already shown a potential for increased usage, as evidenced in a study of low-income women in Brazil [25] and another of patients in the USA [26]. Both these studies indicate that the idea of performing self-tests for sexually transmitted infections can be attractive to users irrespective of their economic context. Already, evidence suggests that the use of pregnancy self-tests remains consistent [27] and, more recently, hepatitis C virus self-test studies are reporting high usability and performance [28,29]. These examples indicate an increasing acceptance of the use of self-tests for identifying the presence of widespread health conditions. The potential for a similar result in the use of HIVST is encouraging, given that in our investigation, an overwhelming percentage of users (above 96%) indicated in the post-test questionnaire that they would use the test again and would recommend it to others. If this is the case, then the oral HIVST has the potential to speed up testing behaviour and balloon the number of people aware of their HIV status. In addition, the large majority indicating that they would prefer to perform the HIVST at home, implies that HIVST can be an attractive way to reach large numbers of the population who may be anxious about undergoing initial tests in more public facilities such as clinics [30].

### Limitations

There were some limitations in this study. The sample selection, while inclusive of participants of varying education levels, comprised of mostly users who had a minimum grade 8 education (59.2%). Thus, it is uncertain whether uneducated to barely educated sectors of the population would be able to fare as well in completing the steps of the HIVST accurately.

Furthermore, while the participants had to be literate in English, slightly more than half cited English as their preferred language. Additionally, there was no further assessment of their competence in the language besides their understanding of the consent form. It is, therefore, unclear whether every participant had an equal chance to succeed in following the instructions accurately.

Although prior experience with HIV testing using an RDT or having a known HIV status were exclusion criteria, people enrolled in this study could have had prior laboratory HIV testing. Our study did not differentiate between individuals with prior HIV testing experience and those who were testing for the first time. The stress experienced (particularly for first time testers) could have contributed to critical errors observed.

While other similar studies have been conducted, each study used different HIVSTs or multiple HIVSTs. As a result, the usability findings could not be assessed comparatively against any other study, due to the absence of a standardised usability assessment tool. This study used a structured usability assessment questionnaire and a semi-structured post-test questionnaire to gauge how user-friendly the device was.

Currently, there is another approved oral-based HIVST available on the South African market. While all steps were taken to limit the enrolment of participants with prior HIVST experience, there was a low possibility that participants may have interacted with the oral test product that is available for purchase.

Lastly, this study used a convenience sample recruited in one geographical region, in one province, and may be less generalisable to the entire country.

## 5. Conclusions

Inexperienced users of the Asante HIVST demonstrated that the device was user-friendly. Additionally, complementary tests of users’ individual results, conducted by trained health professionals, indicated that the device had a very high accuracy rate. Its availability on the South African market has the potential to rapidly increase the speed and volume of HIV self-testing. The processes and verification methods used in this study could well be adopted in future studies on the efficacy of similar devices and could, additionally, serve as a guideline for the development of similar products.

## Figures and Tables

**Figure 1 diagnostics-11-01816-f001:**
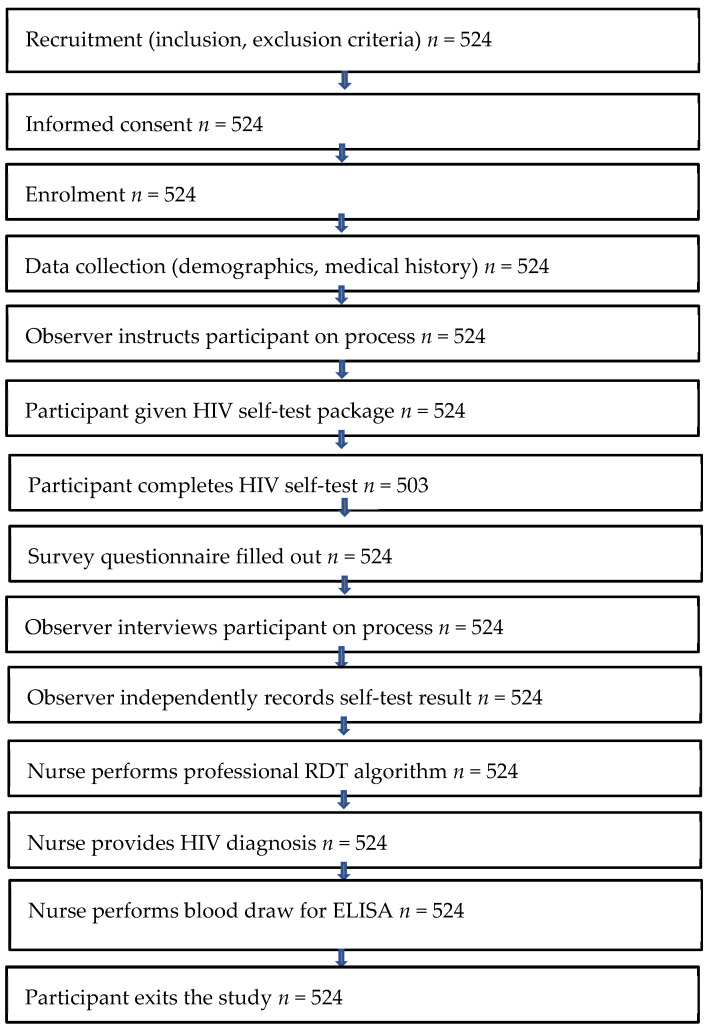
Sample sizes used in calculations.

**Figure 2 diagnostics-11-01816-f002:**
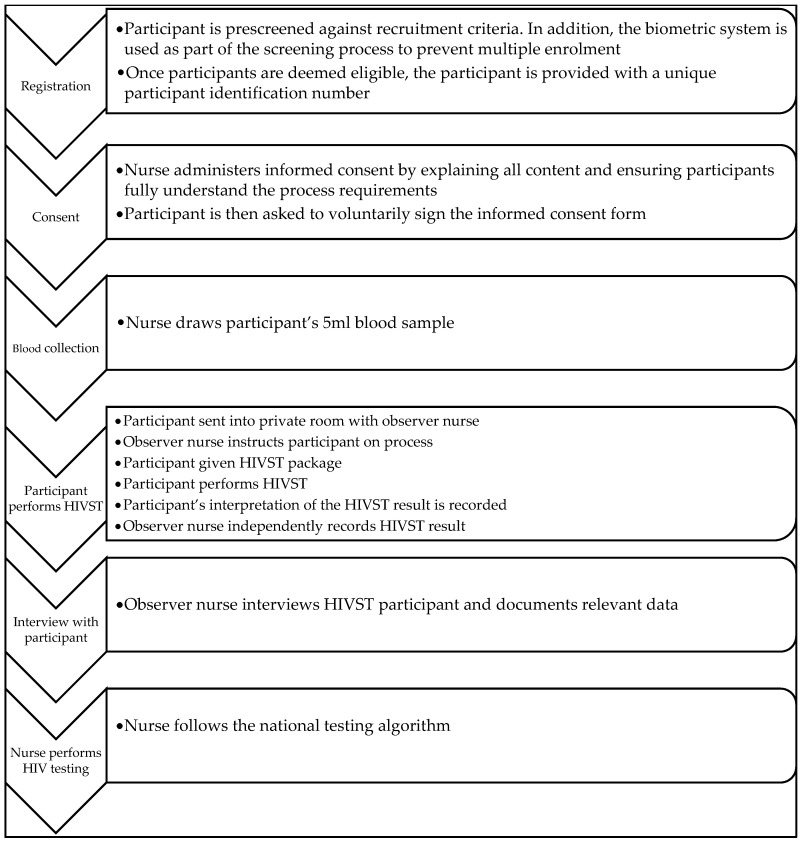
Evaluation study process.

**Figure 3 diagnostics-11-01816-f003:**
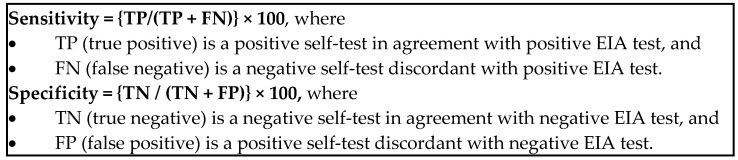
Sensitivity and specificity formulae.

**Table 1 diagnostics-11-01816-t001:** Participant demographic characteristics.

Characteristic	*n* (%)
**Sex**	
Female	276 (52.7%)
Male	248 (47.3%)
**Age**	
18–25	221 (42.2%)
26–35	189 (36.1%)
36–45	81 (15.5%)
46–55	23 (4.4%)
56–65	8 (1.5%)
66+	2 (0.4%)
**Nationality**	
SA	475 (90.6%)
Zimbabwe	43 (8.2%)
Other	6 (1.1%)
**Education level**	
Grade 7 or lower	33 (6.3%)
Grades 8–12	310 (59.2%)
Tertiary	181 (34.5%)
**Employment status**	
Employed	87 (16.6%)
Unemployed	404 (77.1%)
Student	33 (6.3%)

Abbreviation: *n*, number; %, percentage.

**Table 2 diagnostics-11-01816-t002:** Usability Assessment.

Number of Participants Enrolled *n* = 524				
1. Did the study participant read/use the information sheet?	Yes	100.0%	No	0.0%
2. Was it difficult for the study participant to remove the test contents from the box?	No	99.8%	Yes	0.2%
3. Did the study participant have difficulty placing the tube in the tube box?	No	97.3%	Yes	2.7%
4. Did the study participant have difficulty with twisting the cap off the test tube?	No	99.6%	Yes	0.4%
5. Was the study participant able to tear open the swab, packing and remove the swab, holding swab by the handle?	Yes	99.6%	No	0.4%
6. Did the study participant collect the sample correctly (press swab against upper gum line, rub swab head back and forth across the gum)	Yes	98.9%	No	1.1%
7. Did the study participant flip the swab over and repeat for the bottom gum?	Yes	98.7%	No	1.3%
8. Did the study participant successfully hold the tube in the box and press the swab firmly against the tube opening?	Yes	97.1%	No	2.9%
9. Did the study participant gently slide the swab into the tube and press all the way to the bottom?	Yes	95.2%	No	4.8%
10. Did the study participant move the swab up and down at least two times to mix with the fluid, then press the swab against the side of the tube, then remove the swab and discard it?	Yes	96.4%	No	3.2%
11. Did the study participant successfully open the test strip packaging and remove the strip?	Yes	99.2%	No	0.8%
12. Did the study participant successfully place the test strip in the tube with the grey label and arrow pointing down?	Yes	95.6%	No	4.4%
13. Which language was preferred by the user?	English	54.2%		
	Zulu	21.69%		
	Xhosa	8.0%		
	Other	16.2%		
14. What was the participant’s apparent level of stress?	Calm	92.0%		
	Stressed	2.9%		
	Anxious	5.1%		
Average usability score	98.1%			
Average on critical steps	97.0%			

**Table 3 diagnostics-11-01816-t003:** Performance evaluation.

*n* = 524	Asante	Professional RDT 1	Professional RDT 2	ELISA
Negative	432	442	0	439
Positive	73	82	82	85
Exclusions				
Invalid	14	0	0	0
TOTAL	524	524	82	524

Abbreviations: *n*, number; RDT, rapid diagnostic test.

**Table 4 diagnostics-11-01816-t004:** Comparative test results of HIVST, ELISA and RDT Algorithm for RDT 1 and 2.

*n* = 524		Confirmatory Test (ELISA)		RDT 1 Algorithm		RDT 2 Algorithm	
HIVST		Positive	Negative	Positive	Negative	Positive	Negative
	Positive	72	1	72	1	72	0
	Negative	4	426	3	427	3	0

Abbreviations: *n*, number; RDT, rapid diagnostic test.

**Table 5 diagnostics-11-01816-t005:** Comparison of sensitivity and specificity calculations of user interpretation of ST Results.

	Untrained User	Professional User
True positive	72	74
False negative	4	2
True negative	426	425
False positive	1	1
Sensitivity	94.7%	97.4%
Specificity	99.8%	99.8%

Abbreviation: %, percentage.

**Table 6 diagnostics-11-01816-t006:** Post-Test Questionnaire.

Questions	Responses (*n* = 524)					
1. Did you use the instructions sheet?	Yes	100.0%	No	0.0%	Other	0.2%
2. Were the instructions easy to follow?	Yes	99.6%	No	0.2%		
3. Which of the pictures worked well or were not good?	Qualitative question—responses described in text					
4. Was the device easy to use?	Yes	99.4%	No	0.6%		
5. Were you confident with performing this test on your own?	Yes	99.0%	No	1.0%		
6. What should you do if you have a negative result?	Test in 3 months	12.8%	Condom	99.6%	Other	6.7%
7. What should you do if you have a positive result?	Condom	5.3%	Clinic	94.7%		
8. What should you do if you have an invalid result?	Retest	47.9%	Clinic	50.4%	Other	1.7%
9. What should you do if you are not sure of your result?	Retest	7.1%	Clinic	91.8%	Other	1.1%
10. Would you use this test again?	Yes	96.8%	No	3.2%		
11. Would you prefer to use this test at home or get tested at a clinic?	Home	93.1%	Clinic	6.5%	Both	0.4%
12. Would you recommend this test to a sexual partner/friend?	Yes	97.1%	No	2.9%		

Abbreviations: *n*, number.

## Data Availability

The data presented in this study are available on request from the corresponding author. The data are not publicly available to maintain participant confidentiality.

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
