# Peer review of "Assessment of the Sedia HIV Self-Test Device: Usability and Performance in the Hands of Untrained Users in Johannesburg, South Africa"

_diagnostics, 2021, doi:10.3390/diagnostics11101816_

Round 1
Reviewer 1 Report
Accept.
Author Response
Review report 1
English language and style
(x) Moderate English changes required
Thank you. The manuscript has been read through again and revised/corrected where necessary. We have also made revisions in line with Reviewer 2’s comments.
Comments and Suggestions for Authors
Accept.
Review report 2
English language and style
(x) English language and style are fine/minor spell check required
Thank you. The manuscript has been read through again and revised/corrected where necessary. We have also made a few edits (elaboration) to the product descriptions.
Comments and Suggestions for Authors
1) How do you differentiate between an untrained user that hasn't conducted the test vs a user that has previously observed the test being performed? In your study design, there is no differentiation between the two. However, if a sub-group of users that have previously observed these test, they will perform much better and skew the outcomes.
While we understand the point you raise, we are certain that this scenario is minimal and will not bias the data. This was an evaluation of an oral based test of which there currently is only one type in the South African market.
We defined the untrained user as an individual who is not trained in the performance of rapid test diagnostics, or a lay user who has not been provided with any training, support, or guidance on the tests performance other than what is available on the products instruction for use. Each person’s first experience with the test kit was at the point of receipt of the kit for the evaluation. There was no opportunity during the study process to observe any other person (trained or untrained) using this kit.
Further, the exclusion criteria of any previous engagement with an HIV self-testing kit was exercised during recruitment. The research team have been involved in the prequalification processes of the oral test and several blood based self-tests in South Africa. They maintain a database of previous trial participants (verified through biometrics). They are also familiar with the recruitment /catchment areas and do not continuously recruit from the same place to further reduce the chances of recruiting people with prior testing experience.
We had therefore initially never made any distinctions on the two groups of untrained user mentioned by the reviewer as it is not applicable. However, we have noted the small probability of prior exposure in the limitations section as follows:
There currently is another approved oral based HIVST in the South African market. While all steps were taken to limit the enrolment of participants with prior HIVST experience, there was a low possibility that participants may have interacted with the oral test product which is available for purchase.
2) Were the study participants compensated for their time? There is no mention of it in the manuscript.
Yes. The participants were reimbursed R150 for their time. This has been included at the end of section 2.4.
3) I understand that the study is focusing on the accuracy of HIV self-test by untrained users, but how was the psychological stress during the test contributing to the errors. Did you differentiate between an individual that was getting their first HIV test vs an individual that had prior experience with getting tested for HIV?
We did not differentiate between groups with prior HIV testing experience and those without. Prior testing using RDTs (irrespective of outcome) was an exclusion criteria as well as having a known HIV positive status. We acknowledge that there still could be a group who had non-RDT HIV tests. In our study majority (92%) were reported as being calm (observed) and 5.1% anxious (observed). Although the average score on the critical steps was 97%, we have mentioned a limitation about critical errors possibly being linked to stress levels.
Minor Points:
1) Define IFU at the first instance.
This has been edited in line 63.
2) Figure 1 (typo: change foe to for): Nurse performs blood draw "for" ELISA
Thank you. This has been edited.
3) Page 4, Line 130: Spelling (change "enrolment" to "enrollment")
We have maintained British spelling throughout the manuscript and have not made this suggested edit.
4) Erroneous Statement (re-write and match with your data): This yielded a 94.7% sensitivity and a 99.8% sensitivity for the participants interpretation of the self-test against the ELISA 234 result, and a 97.4% sensitivity and 99.8% specificity for professional user interpretation of 235 the self-test against the ELISA.
The sentence has been corrected as follows:
This yielded a 94.7% sensitivity and a 99.8% specificity for the participants’ interpretation of the self-test against the ELISA result, and a 97.4% sensitivity and 99.8% specificity for professional user interpretation of the self-test against the ELISA (Table 4).
Reviewer 2 Report
1) How do you differentiate between an untrained user that hasn't conducted the test vs a user that has previously observed the test being performed? In your study design, there is no differentiation between the two. However, if a sub-group of users that have previously observed these test, they will perform much better and skew the outcomes.
2) Were the study participants compensated for their time? There is no mention of it in the manuscript.
3) I understand that the study is focusing on the accuracy of HIV self-test by untrained users, but how was the psychological stress during the test contributing to the errors. Did you differentiate between an individual that was getting their first HIV test vs an individual that had prior experience with getting tested for HIV?
Minor Points:
1) Define IFU at the first instance.
2) Figure 1 (typo: change foe to for): Nurse performs blood draw "for" ELISA
3) Page 4, Line 130: Spelling (change "enrolment" to "enrollment"
4) Erroroneous Statement (re-write and match with your data): This yielded a 94.7% sensitivity 233 and a 99.8% sensitivity for the participants interpretation of the self-test against the ELISA 234 result, and a 97.4% sensitivity and 99.8% specificity for professional user interpretation of 235 the self-test against the ELISA.
Author Response
Review report 2
English language and style
(x) English language and style are fine/minor spell check required
Thank you. The manuscript has been read through again and revised/corrected where necessary. We have also made a few edits (elaboration) to the product descriptions.
Comments and Suggestions for Authors
1) How do you differentiate between an untrained user that hasn't conducted the test vs a user that has previously observed the test being performed? In your study design, there is no differentiation between the two. However, if a sub-group of users that have previously observed these test, they will perform much better and skew the outcomes.
While we understand the point you raise, we are certain that this scenario is minimal and will not bias the data. This was an evaluation of an oral based test of which there currently is only one type in the South African market.
We defined the untrained user as an individual who is not trained in the performance of rapid test diagnostics, or a lay user who has not been provided with any training, support, or guidance on the tests performance other than what is available on the products instruction for use. Each person’s first experience with the test kit was at the point of receipt of the kit for the evaluation. There was no opportunity during the study process to observe any other person (trained or untrained) using this kit.
Further, the exclusion criteria of any previous engagement with an HIV self-testing kit was exercised during recruitment. The research team have been involved in the prequalification processes of the oral test and several blood based self-tests in South Africa. They maintain a database of previous trial participants (verified through biometrics). They are also familiar with the recruitment /catchment areas and do not continuously recruit from the same place to further reduce the chances of recruiting people with prior testing experience.
We had therefore initially never made any distinctions on the two groups of untrained user mentioned by the reviewer as it is not applicable. However, we have noted the small probability of prior exposure in the limitations section as follows:
There currently is another approved oral based HIVST in the South African market. While all steps were taken to limit the enrolment of participants with prior HIVST experience, there was a low possibility that participants may have interacted with the oral test product which is available for purchase.
2) Were the study participants compensated for their time? There is no mention of it in the manuscript.
Yes. The participants were reimbursed R150 for their time. This has been included at the end of section 2.4.
3) I understand that the study is focusing on the accuracy of HIV self-test by untrained users, but how was the psychological stress during the test contributing to the errors. Did you differentiate between an individual that was getting their first HIV test vs an individual that had prior experience with getting tested for HIV?
We did not differentiate between groups with prior HIV testing experience and those without. Prior testing using RDTs (irrespective of outcome) was an exclusion criteria as well as having a known HIV positive status. We acknowledge that there still could be a group who had non-RDT HIV tests. In our study majority (92%) were reported as being calm (observed) and 5.1% anxious (observed). Although the average score on the critical steps was 97%, we have mentioned a limitation about critical errors possibly being linked to stress levels.
Minor Points:
1) Define IFU at the first instance.
This has been edited in line 63.
2) Figure 1 (typo: change foe to for): Nurse performs blood draw "for" ELISA
Thank you. This has been edited.
3) Page 4, Line 130: Spelling (change "enrolment" to "enrollment")
We have maintained British spelling throughout the manuscript and have not made this suggested edit.
4) Erroneous Statement (re-write and match with your data): This yielded a 94.7% sensitivity and a 99.8% sensitivity for the participants interpretation of the self-test against the ELISA 234 result, and a 97.4% sensitivity and 99.8% specificity for professional user interpretation of 235 the self-test against the ELISA.
The sentence has been corrected as follows:
This yielded a 94.7% sensitivity and a 99.8% specificity for the participants’ interpretation of the self-test against the ELISA result, and a 97.4% sensitivity and 99.8% specificity for professional user interpretation of the self-test against the ELISA (Table 4).